# Condense, Don't Just Prune: Enhancing Efficiency and Performance in MoE Layer Pruning

**Mingyu Cao**                                                    *m.cao@surrey.ac.uk*
*University of Surrey*

**Gen Li**                                                         *gen@g.clemson.edu*
*Clemson University*

**Jie Ji**                                                         *jji@g.clemson.edu*
*Clemson University*

**Jiaqi Zhang**                                          *zhangjiaqi39@meituan.com*
*Meituan*

**Ajay Jaiswal**                                           *ajayjaiswal@utexas.edu*
*University of Texas at Austin*

**Li Shen**                                                *mathshenli@gmail.com*
*Sun Yat-sen University*

**Xiaolong Ma**                                            *xiaolongma@arizona.edu*
*The University of Arizona*

**Shiwei Liu**                                                 *sliu@tue.ellis.eu*
*ELLIS Institute Tübingen, Max Planck Institute for Intelligent Systems, Tübingen AI Center*

**Lu Yin** [*]                                                   *l.yin@surrey.ac.uk*
*University of Surrey*

**Reviewed on OpenReview:** *https://openreview.net/forum?id=BQe6j6sAu6*

## Abstract

Mixture-of-Experts (MoE) has garnered significant attention for its ability to scale up neural networks while utilizing the same or even fewer active parameters. However, MoE does not relieve the massive memory requirements of networks, which limits their practicality in real-world applications, especially in the era of large language models (LLMs). While recent work explores the possibility of removing entire layers of MoE to reduce memory, the performance degradation is still notable. In this paper, we propose ConDense-MoE (CD-MoE) that, instead of dropping the entire MoE layer, condenses the large, sparse MoE layer into a smaller, denser layer with only a few experts activated for all tokens, while maintaining hardware friendliness. Our approach is specifically designed for fine-grained MoE with shared experts, where Feed-Forward Networks are split into many small experts, with certain experts isolated to serve as shared experts that are always activated, such as DeepSeekMoE and QwenMoE. We demonstrate the effectiveness of our method. Specifically, for the DeepSeekMoE-16B model, our approach maintains **90%** of the average accuracy while reducing memory usage by **27.5%** and increasing inference speed to **1.26** times. Moreover, we show that by applying lightweight expert fine-tuning—only to the condensed layers—and using 5 hours on a single 80G A100 GPU, we can successfully recover **98%** of the original performance. Our code is available at: `https://github.com/duterscmy/CD-MoE`.

---

[*]Corresponding author.

# 1  Introduction

Large Language Models (LLMs) continue to grow in both size and capability (Brown, 2020; Touvron et al., 2023b;a), fueling demand for architectures that can scale with minimal increases in computational cost. Mixture of Experts (MoE) architectures have emerged as a promising solution to this challenge (Shazeer et al., 2017; Jiang et al., 2024; Team et al., 2024). Unlike standard dense networks, MoEs selectively activate only the most relevant subset of parameters (referred to as "experts") for a given input, enabling substantial growth in model capacity without a proportional escalation in compute overhead. Recent advances in fine-grained expert segmentation (Dai et al., 2024; Team, 2024) push this concept further by splitting feed-forward layers into many small experts, while designating a small set of shared experts that remain active across all tokens. Despite the clear benefits, MoE architectures still face major deployment barriers due to high memory costs. Storing a large number of experts, even if most remain dormant per token, can be prohibitively expensive in real-world systems. This reality highlights the urgent need for approaches that preserve the performance benefits of MoEs while lowering their memory footprint.

In this paper, we present a new framework called **ConDense-MoE** (`CD-MoE`), a novel framework that significantly enhances the efficacy specifically for DeepSeek-style shared-expert-type MoE (Dai et al., 2024; Team, 2024). Unlike prior MoE compression approaches that often remove entire layers or retain the routing mechanism with fewer experts (He et al., 2024b), our method eliminates the routing process at selected layers and condenses them into dense layers. Specifically, we devise a greedy strategy to identify which layers—and which experts within them—can be condensed with minimal impact on model accuracy, leading to significant reductions in memory usage and faster inference.

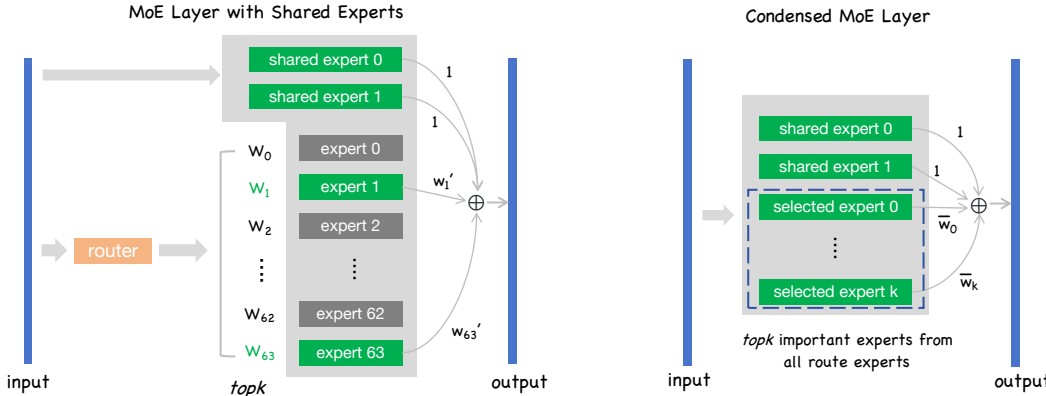

Figure 1: **Left:** The structure of the Deepseek MoE layer. $w_i'$ represents the weights after normalization. **Right:** The structure of the ConDense-MoE layer, where only the most important top-k experts are retained. $\bar{w}_i$ represents the fixed weights that are pre-computed during condensing using the average weight of all calibration tokens.

`CD-MoE` stems from a surprising yet critical insight: in shared-expert–type MoEs, simply removing the routing mechanism while retaining only the shared experts at a given layer generally incurs minor output difference. Building on this, `CD-MoE` preserves these essential shared experts and a small subset of routing experts, thereby condensing originally sparse layers into compact, dense analogs. As these preserved experts account for only a small fraction of overall parameters, we substantially cut memory requirements without sacrificing much performance. Our experiments demonstrate that on *DeepSeekMoE-16B* (Dai et al., 2024), `CD-MoE` reduces memory by 27.5% and maintains nearly 90% of the original accuracy on a suite of downstream tasks. Moreover, our method uncovers additional experts that are beneficial for fine-tuning, enabling up to 98% recovery of the original performance after a lightweight supervised fine-tuning that updates only the condensed layers. In summary, our contributions are:

- **A novel MoE condensation approach** that selectively removes less critical experts and condenses large sparse MoE layers into small, dense structures.

- **An efficient selection algorithm** that identifies which layers to condense and which experts to preserve, achieving a favorable trade-off between memory savings and accuracy.

- **Empirical validation** on *DeepSeekMoE-16B*, demonstrating a 27.5% memory reduction and a 1.26× inference speed, while retaining up to 90% of the zero-shot accuracy. Moreover, lightweight fine-tuning can effectively reclaim 98% of the original performance using only a single A100 GPU in a few hours.

## 2 Related Work

Mixture-of-Experts (MoE) architectures have become a powerful framework for scaling neural networks without linearly increasing computational burden (Shazeer et al., 2017; Jiang et al., 2024; Dai et al., 2024; Team, 2024). Early methods such as Switch Transformer (Fedus et al., 2022) demonstrated that sparse activation can substantially boost model capacity while maintaining tractable compute. Further work has refined routing policies (Jiang et al., 2024) and introduced a fine-grained segmentation of feed-forward layers into many small experts, often augmented with "shared" experts that remain active for every token (Dai et al., 2024; Team, 2024). These advances collectively enhance training stability and generalization, yet the growing pool of experts poses serious deployment challenges, chiefly due to large memory demands.

A range of pruning and compression strategies have emerged to address these overheads. Some works prune experts by identifying those that are minimally activated or less essential for downstream tasks (Chi et al., 2022; Sun et al., 2024; Muzio et al., 2024), with solutions ranging from exhaustive searches over all expert subsets (Lu et al., 2024) to filtering experts by their activation frequency. While these methods typically preserve routing, more aggressive approaches prune entire MoE layers, achieving considerable memory reductions but at the risk of pronounced accuracy losses (He et al., 2024b). In contrast, our technique selectively eliminates the routing mechanism in particular layers and retains only a small subset of experts (including the shared experts), thus reducing memory demands while sustaining most of the model's representational power.

## 3 ConDense-MoE (`CD-MoE`)

In this section, we present a comprehensive explanation of the `CD-MoE` process, which follows a sequential workflow comprising several critical steps. First, we delve into *Expert Selection and Condensing* (Section 3.2), where we detail the methods used to identify and condense the most essential experts from all available experts within a layer. Second, we explore *Layer Selection and Condensing* (Section 3.3), focusing on selecting the most suitable layers for condensation to preserve as much of the LLM's inherent capabilities as possible. Last but not least, we highlight that only performing *lightweight fine-tuning* (Section 4.3) on condensed layers, can almost recover the original performance of MoE.

### 3.1 Preliminaries of MoE

Here we define the basic structure of an MoE layer. All experts in an MoE layer are represented as $\{E_1, E_2, \ldots, E_n\}$. For most fine-grained MoE models, in addition to routing experts, the shared experts $E_s$ is used. Any token will activate the shared experts and several routing experts, as shown in Figure 1 left. For the input data $X$, the output is computed as follows:

$$\mathbf{h}_t = \sum_{i=1}^{N} \left( g_{i,t} E_i \left( \mathbf{x}_t \right) \right) + E_s \left( \mathbf{x}_t \right) + \mathbf{x}_t, \tag{1}$$

where

$$g_{i,t} = \begin{cases} s_{i,t}, & s_{i,t} \in \mathrm{TopK} \left( \{ s_{j,t} \mid 1 \leq j \leq N \}, k \right), \\ 0, & \text{otherwise}, \end{cases}$$

$$s_{i,t} = \mathrm{Softmax}_i \left( \mathbf{x}_t^T \mathbf{e}_i \right).$$

Here, $N$ denotes the total number of route experts, $E_i(\cdot)$ represents the $i$-th expert, $E_s$ is the shared expert, $g_{i,t}$ is the gate value for the $i$-th expert, $s_{i,t}$ denotes the token-to-expert similarity, $\mathrm{TopK}(\cdot, k)$ represents the

set consisting of the $k$ highest similarity scores among those computed for the $t$-th token and all $N$ experts, and $\mathbf{e}_i$ is the centroid of the $i$-th expert.

## 3.2 Expert Selection and Condensing

---

**Algorithm 1:** Condense Experts Selection

---

Input: Calibration data input $X$, routing expert set $E_{\text{route}} = \{E_1, \ldots, E_n\}$, number of selected experts $K$

Output: Condense expert set $E_{\text{condense}} = \{E_{\text{shared}}\}$

**Initialize** $E_{\text{condense}} \leftarrow \{E_{\text{shared}}\}$, $O_{\text{ref}} \leftarrow \text{Layer}_{\text{route}}(X)$ ▷ //Compute reference output with all routing experts

for $k \leftarrow 1$ **to** $K$ do

    for each expert $E_i \in E_{\text{route}}$ do

        $E_{\text{condense}} \leftarrow E_{\text{condense}} \cup \{E_i\}$ ▷ //Temporarily add expert to condense set

        $O_{\text{tmp}} \leftarrow \text{Layer}_{\text{condense}}(X)$ ▷ //Compute output with current condense experts

        $loss_i \leftarrow \text{JS}(O_{\text{ref}}, O_{\text{tmp}})$ ▷ //Compute JS divergence as loss for $E_i$

        $E_{\text{condense}} \leftarrow E_{\text{condense}} \setminus \{E_i\}$ ▷ //Remove expert after computing loss

    $E_{\text{best}} \leftarrow \text{argmin}_{E_i \in E_{\text{route}}} loss_i$ ▷ //Select the expert with minimum loss

    $E_{\text{condense}} \leftarrow E_{\text{condense}} \cup \{E_{\text{best}}\}$ ▷ //Add best expert to condense set

    $E_{\text{route}} \leftarrow E_{\text{route}} \setminus \{E_{\text{best}}\}$ ▷ //Remove selected expert from routing set

---

Our `CD-MoE` framework removes the standard routing mechanism during inference. For each expert $E_i$, we use open-source C4 calibration data to compute the mean of $g_{i,t}$ when that expert is activated, which is then used as the fixed gate value $g_i$ for that expert during the inference phase:

$$g_i = (1/|t \mid g_{i,t} \neq 0|) \sum_{t \mid g_{i,t} \neq 0} g_{i,t} \tag{2}$$

For the condense layer, we only retain $K$ routing experts including the shared experts, which is consistent with the number of active experts during the training phase. Additionally, benefiting from sufficient training during the training stage, the shared expert is retained, as shown in Figure 1 right. After condensation, the MoE layer is represented as:

$$\mathbf{h}_t = \sum_{i=1}^{K} (g_{i,t} E_i (\mathbf{x}_t)) + E_s (\mathbf{x}_t) + \mathbf{x}_t \tag{3}$$

Next, we describe how to select the most crucial experts. Lu et al. (2024) explore all possible expert combinations and choose the combination of experts that makes the layer outputs before and after pruning as close as possible, while pruning the rest. However, for fine-grained MoE models like *DeepSeekMoE-16B* (Dai et al., 2024) and *Qwen1.5-MoE-A2.7B* (Team, 2024), where the number of experts often exceeds 60, testing all permutations of experts requires significant computational resources and time. Therefore, we propose using a greedy search approach Algorithm 1 to select the most critical experts.

In which, $K$ is consistent with the number of routing experts(non-shared experts) activated in the original model, and $JS(.)$ is the Jensen-Shannon divergence used to measure the difference between two outputs. Zhang et al. (2024) shown that in model pruning, this metric is better than angular distance and Euclidean distance. We also demonstrated its superiority over other metrics such as Kullback–Leibler ($KL$) divergence and perplexity in Table 8.

## 3.3 Layer Selection and Condensing

Clark et al. (2019b) have confirmed that different layers in language models exhibit distinct functionalities. While our objective is to retain the most impactful experts during the condensing process of each layer, it is

---

**Algorithm 2:** Condense Layers Selection

---

Input: Data input $X$, routing layer set $L_{\text{route}} = \{L_1, \ldots, L_n\}$, number of condensed layers $N$

Output: Condense layer set $L_{\text{condense}} = \{\}$

**Initialize** $L_{\text{condense}} \leftarrow \{\}$, $O_{\text{ref}} \leftarrow \text{Model}(X)$ ▷ //Compute reference output with all routing layers

for $n \leftarrow 1$ **to** $N$ do

    for each layer $L_i \in L_{\text{route}}$ do

        $L_{\text{condense}} \leftarrow L_{\text{condense}} \cup \{L_i\}$ ▷ //Temporarily add layer to condense set

        $O_{\text{tmp}} \leftarrow \text{Model}(X)$ ▷ //Compute output with current condense layers

        $loss_i \leftarrow \text{JS}(O_{\text{ref}}, O_{\text{tmp}})$ ▷ //Compute JS divergence as loss for $L_i$

        $L_{\text{condense}} \leftarrow L_{\text{condense}} \setminus \{L_i\}$ ▷ //Remove layer after computing loss

    $L_{\text{best}} \leftarrow \text{argmin}_{L_i \in L_{\text{route}}} loss_i$ ▷ //Select the layer with minimum loss

    $L_{\text{condense}} \leftarrow L_{\text{condense}} \cup \{L_{\text{best}}\}$ ▷ //Add best layer to condense set

    $L_{\text{route}} \leftarrow L_{\text{route}} \setminus \{L_{\text{best}}\}$ ▷ //Remove selected layer from routing set

---

inevitable that the outputs will deviate from their original values due to reduced capacity. Indiscriminately condensing all layers can thus lead to significant degradation in model performance and the loss of essential features. Therefore, akin to our approach in expert selection, we prioritize condensing those layers that exert minimal impact on model output changes post-pruning.

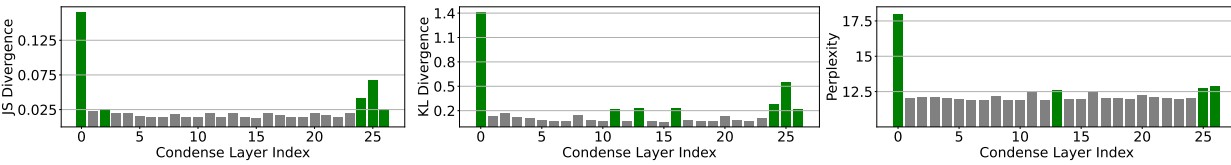

Figure 2: Left: Fluctuations in the $JS$ divergence between the the outputs of the condensed model and the original dense model across different layers. Middle: fluctuations in the $KL$ divergence. Right: fluctuations in the perplexity.

To systematically quantify the impact of condensing individual layers, we conducted preliminary experiments assessing the output changes using the $JS$ divergence, $KL$ divergence and perplexity between the pruned and unpruned model outputs as our evaluation metric. As illustrated in Figure 2, condensing layer 0 results in a significant shift in the output distribution, indicating its critical role in the model's performance. While the middle layers exhibit some fluctuation, condensing the last three layers also causes noticeable deviations, suggesting that these layers are essential for maintaining output fidelity. These observations indicate that condensing different layers introduces varying degrees of change in the three metrics, reflecting their varying importance to the overall model performance.

Motivated by these findings, we adopt a greedy search strategy analogous to that used in expert selection to determine the optimal layers for condensing. Specifically, we compute the $JS$ divergence before and after condensing based on the output of the final layer, iteratively adding layers to the $Layer_{condense}$ set that can minimize the distance between the model's final layer output and the output before condensing, as detailed in Algorithm 2. For layers that are not condensed, the standard token routing mechanisms are employed during forward propagation to maintain their full expressive power. For the layers being condensed, the routing mechanism is removed, and the MOE is compressed into a dense structure. The number of condensing layers **N** in can be chosen based on goals (e.g., memory budget vs. desired accuracy). This selective condensing approach allows us to effectively reduce computational complexity and memory usage while minimizing degradation in model performance.

Table 1: Results on *DeepSeekMoE-16B*. The underlined variables are tried to be kept consistent for fair comparison, and the **bolds** represent the best results. '#*L*' represents the number of pruned layers; 'ACT.' represents the number of activated parameters; and 'MEM.' is the remaining memory ratio compared to the original model. The lightweight SFT results are averaged over three runs with different random seeds, and the standard deviation is indicated below the accuracy.

| method | #L | ACT. | MEM. | Speedup | ARC-C | BOOLQ | HELLA | MMLU | OBQA | PIQA | RTE | WINO | **AVG.** |
|---|---|---|---|---|---|---|---|---|---|---|---|---|---|
| GPT-NEO-2.7B | - | 2.7B | - | - | 29.8 | 61.7 | 55.2 | 25.0 | 34.6 | 72.9 | 53.4 | 57.8 | 48.8 |
| OPT-2.7B | - | 2.7B | - | - | 31.2 | 59.9 | 60.6 | 25.4 | 35.2 | 74.7 | 54.2 | 60.5 | 50.2 |
| BLOOM-3B | - | 3B | - | - | 30.4 | 61.4 | 54.5 | 25.9 | 32.4 | 70.7 | 56.0 | 58.6 | 48.7 |
| OPENLLAMA-3B | - | 3B | - | - | 36.1 | 67.1 | 64.4 | 23.9 | 38.6 | 75.1 | 54.2 | 62.3 | 52.7 |
| UNPRUNED | - | 2.8B | 100% | 1.0x | 48.3 | 72.7 | 77.4 | 38.3 | 44.2 | 78.7 | 63.2 | 70.1 | 61.6 |
| | | | | | W/O SFT | | | | | | | | |
| LAYERDROP | 8 | 2.4B | 72.2% | 1.34x | 35.3 | 65.4 | 57.5 | 29.8 | 32.6 | 62.9 | 48.0 | 63.6 | 49.4 |
| BLOCKDROP | 8 | 2.3B | 71.3% | 1.42x | 36.2 | 62.2 | 57.2 | 29.0 | 34.6 | 70.2 | 63.2 | 64.7 | 52.2 |
| M-SMoE | 9 | 2.8B | 70.0% | 1.0x | 37.1 | 71.1 | 64.0 | 28.0 | 36.4 | 75.5 | 59.9 | 67.2 | 54.9 |
| CD-MoE-S | 8 | 2.4B | 73.1% | 1.34x | 37.8 | 70.6 | 65.9 | 28.8 | 38.6 | 75.0 | 58.8 | 65.0 | 55.1 |
| CD-MoE-SR | 9 | 2.8B | 72.5% | 1.26x | 37.9 | 72.3 | 64.4 | 27.1 | 37.6 | 75.9 | 61.7 | 67.0 | **55.5** |
| | | | | | W/ LIGHTWEIGHT SFT | | | | | | | | |
| LAYERDROP | 8 | 2.4B | 72.2% | 1.34x | 37.8 | 71.0 | 58.9 | 39.0 | 34.2 | 64.5 | 60.8 | 70.0 | 54.5 |
| | | | | | ±0.7 | ±1.0 | ±0.4 | ±0.7 | ±0.9 | ±1.1 | ±0.4 | ±0.3 | ±0.3 |
| CD-MoE-S | 8 | 2.4B | 73.1% | 1.34x | 41.5 | 72.0 | 66.0 | 39.5 | 40.3 | 75.9 | 67.7 | 70.0 | 59.1 |
| | | | | | ±0.7 | ±0.5 | ±0.5 | ±0.8 | ±0.5 | ±0.7 | ±1.1 | ±0.4 | ±0.2 |
| CD-MoE-SR | 9 | 2.8B | 72.5% | 1.26x | 42.4 | 79.4 | 66.7 | 38.5 | 38.6 | 76.1 | 67.0 | 73.9 | **60.4** |
| | | | | | ±0.6 | ±0.8 | ±0.3 | ±0.7 | ±0.8 | ±0.7 | ±0.3 | ±1.5 | ±0.2 |

### 3.4 Time Complexity Analysis

The computational efficiency of our greedy search algorithms stems from their polynomial time complexity relative to key model parameters:

- **Greedy Expert Selection:** For a layer with $E$ routing experts and target selection of $K$ routing experts, each iteration evaluates all remaining candidates . The total complexity is $O(K \cdot E)$, which improves upon exhaustive search's $O(2^E)$ complexity (Lu et al., 2024).

- **Greedy Layer Selection:** With $L$ candidate layers and $N$ target condensed layers, each step evaluates up to $L$ layers. The complexity becomes $O(N \cdot L)$.

Taking *DeepSeekMoE-16B* ($E = 66$, $L = 27$) as an example, selecting $K = 6$ routing experts per condensed layer requires only 369 inferences, demonstrating practical scalability. The linear dependence on $E$ and $L$ makes our approach feasible for billion-parameter MoEs.

## 4 Experiments

### 4.1 Experiment Setup

We adopt the open-source models *DeepSeekMoE-16B* and *Qwen1.5-MoE-A2.7B* (Team, 2024) (with 16B and 14.3B parameters, respectively) as representative high-capacity examples, and a larger model *Qwen2-MoE-57B-A14B*. We randomly sample 100 examples from the open-source C4 dataset to serve as calibration data during the condensing phase of our model, due to its widespread adoption in the LLM community (Sun et al., 2024; Frantar & Alistarh, 2023; Yin et al., 2024).

**Evaluation**: To provide a comprehensive assessment of our model's performance, we follow the protocol established by He et al. (2024b). We report zero-shot accuracies on eight diverse tasks selected from

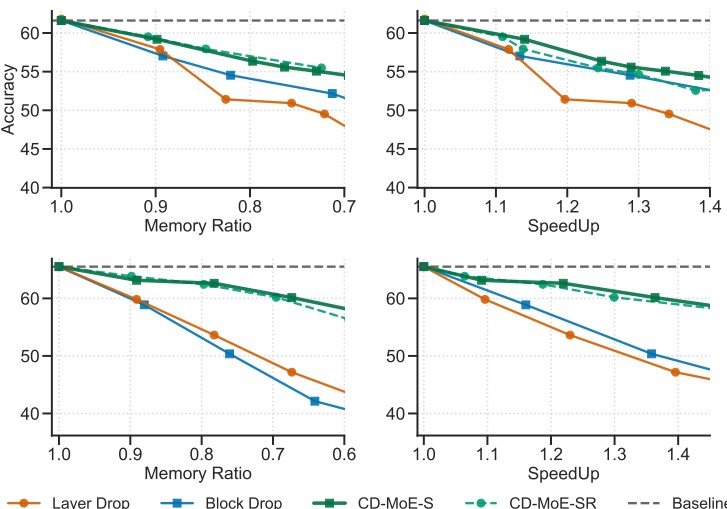

Figure 3: `CD-MoE` against baselines on zero-shot tasks w/o fine-tuning on *DeepSeekMoE-16B* (top) and *Qwen1.5-MoE-A2.7B* (bottom). Left: Average accuracy with varying Memory Ratio against the original model. Right: Average accuracy with varying SpeedUp against the original model. The gray dotted line is the original model result.

the EleutherAI Language Model Evaluation Harness (Gao et al., 2024), which includes the 7 open-source commonsense reasoning benchmark datasets: *ARC-Challenge* (Clark et al., 2018), *BoolQ* (Clark et al., 2019a), *HellaSwag* (Zellers et al., 2019), *OpenBookQA* (Mihaylov et al., 2018), *PIQA* (Bisk et al., 2019), *RTE* (Dagan et al., 2005), *WinoGrande* (Sakaguchi et al., 2019), and a more challenging dataset *MMLU* (Hendrycks et al., 2021). In addition, we evaluate the reasoning ability of the models on the mathematical task *GSM8K* (Cobbe et al., 2021) and the code generation task *HumanEval* (Chen et al., 2021).

For quality assessment, we use accuracy as the evaluation metric for commonsense reasoning, *MMLU*, and *GSM8K*. For *HumanEval*, we adopt Pass@1 as the metric. To provide a more detailed comparison across different pruning methods, we employ the following two metrics:

- **SpeedUp**: We perform inference 5,000 times on a fixed input sequence and measure the average latency. SpeedUp is computed as $\text{SpeedUp} = \frac{\text{Latency(Original Model)}}{\text{Latency(Pruned Model)}}$.

- **Memory Ratio**: We measure the peak GPU memory consumption during inference for the full model and each pruned variant using a fixed input. Memory Ratio is computed as $\text{Memory Ratio} = \frac{\text{Peak Memory(Pruned Model)}}{\text{Peak Memory(Original Model)}}$.

Additionally, the Recovery Ratio reported in our abstract and experiment analysis is computed as: $\frac{\text{Average Accuracy(Pruned Model)}}{\text{Average Accuracy(Original Model)}}$.

**Experimental Methods**: We primarily compare our method against the *Block Drop* and *Layer Drop* (He et al., 2024b) and *M-SMoE* (Li et al., 2024b). These methods serve as strong baselines for pruning MoE models and are directly relevant to our study. *Block Drop* and *Layer Drop* **prune all experts** and the routing process in the layer, which is more aggressive than our method, while *M-SMoE* more conservatively retains the routing process and only merges the experts into a smaller number. We also compared the results with dense models that have similar activations, such as GPT-Neo-2.7B (Gao et al., 2020), OPT-2.7B (Zhang et al., 2022), BLOOM-3B (Press et al., 2022), and OpenLLaMA-3B (Touvron et al., 2023a).

For our proposed method, we introduce two variants::

- `CD-MoE-S`: This variant retains only the **S**hared experts after condensation. The number of shared experts varies across different models, as detailed in Appendix A.1.

Table 2: Results on *Qwen1.5-MoE-A2.7B*. The underlined variables are tried to be kept consistent for fair comparison, and the **bolds** represent the best results. '#*L*' represents the number of pruned layers; 'ACT.' represents the number of activated parameters; and 'MEM.' is the remaining memory ratio compared to the original model.

| method | #L | ACT. | MEM. | Speedup | ARC-C | BOOLQ | HELLA | MMLU | OBQA | PIQA | RTE | WINO | **AVG.** |
|---|---|---|---|---|---|---|---|---|---|---|---|---|---|
| UNPRUNED | | 2.7B | 100% | 1.0x | 45.0 | 79.5 | 77.3 | 61.2 | 43.6 | 80.3 | 67.9 | 69.3 | 65.5 |
| | | | | | W/O SFT | | | | | | | | |
| LAYERDROP | 6 | 2.3B | 78.3% | 1.23x | 34.4 | 62.6 | 63.1 | 45.4 | 32.8 | 73.6 | 54.5 | 62.6 | 53.6 |
| BLOCKDROP | 6 | 2.2B | 76.1% | 1.36x | 31.3 | 62.2 | 57.4 | 38.4 | 32.0 | 70.7 | 52.7 | 58.2 | 50.4 |
| MC-SMoE | 6 | 2.7B | 73.0% | 1.0x | 40.9 | 76.1 | 71.5 | 53.2 | 39.7 | 77.5 | 70.5 | 64.7 | 61.8 |
| CD-MoE-S | 6 | 2.5B | 78.3% | 1.22x | 41.1 | 75.8 | 72.9 | 56.0 | 41.0 | 78.9 | 67.2 | 68.1 | **62.6** |
| CD-MoE-SR | 6 | 2.7B | 79.7% | 1.19x | 40.2 | 77.2 | 72.1 | 54.9 | 39.4 | 78.0 | 71.1 | 66.4 | 62.4 |

Table 3: Results on *Qwen2-MoE-57B-A14B*. The notations (#L, ACT., MEM.) and formatting conventions follow Table 2.

| method | #L | ACT. | MEM. | Speedup | ARC-C | BOOLQ | HELLA | OBQA | PIQA | RTE | **AVG.** |
|---|---|---|---|---|---|---|---|---|---|---|---|
| UNPRUNED | - | 14.2B | 100% | 1.0x | 55.6 | 88.2 | 82.3 | 45.6 | 82.0 | 76.9 | 71.8 |
| | | | | | W/O SFT | | | | | | |
| LAYERDROP | 6 | 11.6B | 79.3% | 1.24x | 42.9 | 77.3 | 69.0 | 37.6 | 76.0 | 69.7 | 62.1 |
| BLOCKDROP | 6 | 11.4B | 79.0% | 1.35x | 28.3 | 61.3 | 51.6 | 32.6 | 62.7 | 59.9 | 49.7 |
| MC-SMoE | 6 | 14.2B | 76.4% | 1.0x | 53.2 | 87.5 | 77.4 | 42.0 | 80.2 | 79.4 | 70.0 |
| CD-MoE-S | 6 | 12.9B | 81.6% | 1.22x | 54.0 | 86.7 | 79.1 | 43.4 | 81.5 | 78.5 | **70.5** |
| CD-MoE-SR | 6 | 14.2B | 83.9% | 1.20x | 54.1 | 86.6 | 78.6 | 43.0 | 80.7 | 80.2 | **70.5** |

- **CD-MoE-SR**: This variant selects not only the **S**hared expert(s) but also an additional $K$ **R**outed experts, preserving them as dense. Here, $K$ is consistent with the number of routed experts activated in the original model, i.e., $K$ = Number of Activated Experts − Number of Shared Experts (see Appendix A.1).

## 4.2 Zero-Shot Evaluation

To assess the effectiveness of our condensation approach in a purely zero-shot setting, we evaluate `CD-MoE-S` and `CD-MoE-SR` on eight downstream tasks without any additional fine-tuning. Table 1 shows the results on *DeepSeekMoE-16B*, we report each model's performance alongside its memory reduction ratio. Because pruning different layers can lead to varying memory footprints, we normalize the number of pruned layers so that all methods operate under comparable memory budgets.

Specifically, ❶ **Comparison to Baselines:** Under similar memory constraints, both `CD-MoE-S` and `CD-MoE-SR` consistently outperform the baselines—*Block Drop*, *Layer Drop* and *M-SMoE*—highlighting the importance of preserving the most informative MoE experts. Moreover, Figure 3 plots the average accuracy against increasing pruning rates and memory savings, illustrating that the performance gap between `CD-MoE` and conventional pruning techniques widens as pruning becomes more aggressive. ❷ **Comparison to Small Dense LLMs:** Notably, even without fine-tuning, `CD-MoE-S` and `CD-MoE-SR` often surpass smaller dense LLMs of comparable size (e.g., BLOOM-3B, OpenLLaMA-3B). This suggests that training a dense model from scratch at a reduced scale does not match the performance of selectively condensed MoE architectures that retain high-value parameters. ❸ `CD-MoE-S` **vs.** `CD-MoE-SR`: In zero-shot mode, `CD-MoE-S` exhibits comparable or slightly superior performance to `CD-MoE-SR`. However, as detailed in Section 4.3, the routed experts in `CD-MoE-SR` confer a substantial advantage during fine-tuning: they enable nearly 98% of the original performance to be recovered in just a few hours on a single 80GB A100 GPU.

Table 2 and Table 3 presents the results on the *Qwen1.5-MoE-A2.7B* and *Qwen2-MoE-57B-A14B*. Under similar memory costs, the advantages of `CD-MoE` over baseline methods are **more pronounced**, and with

Table 4: Results on *GSM8K* and *HumanEval*. The underlined variables are tried to be kept consistent for fair comparison, and the **bolds** represent the best results. '#*L*' represents the number of pruned layers and 'MEM.' is the remaining memory ratio compared to the original model.

| Method | DeepSeekMoE-16B | | | | | Qwen1.5-MoE-A2.7B | | | | |
|---|---|---|---|---|---|---|---|---|---|---|
| | #L | MEM. | SpeedUp | GSM8K | HumanEval | #L | MEM. | SpeedUp | GSM8K | HumanEval |
| UNPRUNED | 0 | 100% | 1.0x | 63.7 | 4.9 | 0 | 100% | 1.0x | 66.1 | 6.1 |
| LAYER DROP | 9 | 72.20% | 1.34x | 48.1 | 2.4 | 6 | 78.30% | 1.23x | 56.5 | 4.3 |
| BLOCK DROP | 8 | 71.30% | 1.42x | 43.5 | 2.4 | 6 | 76.10% | 1.36x | 52.9 | 2.4 |
| MC-SMoE | 8 | 70.00% | 1.0x | 53.5 | **3.7** | 6 | 73.00% | 1.0x | 57.4 | 4.3 |
| CD-MoE-S | 8 | 73.10% | 1.34x | 52.8 | **3.7** | 6 | 78.30% | 1.22x | 58.1 | **4.9** |
| CD-MoE-SR | 9 | 72.50% | 1.26x | **54.0** | **3.7** | 6 | 79.70% | 1.19x | **58.9** | 4.3 |

more than a 20% reduction in memory usage, the average accuracy approaches that of the original model. We attribute this to the fact that the Qwen series MoE models have a larger proportion of shared experts, as shown in Table 8. `CD-MoE` preserves these experts, enabling the condensed model to retain more knowledge acquired during the pre-training phase.

Table 4 presents the results on *GSM8K* and *HumanEval*. Similar to the observations on commonsense reasoning, `CD-MoE` maintains strong performance under comparable memory footprints. Overall, these results confirm that the condensation strategy preserves core model capacities more effectively than layer-level pruning alone. By selectively retaining expert parameters, `CD-MoE` consistently delivers strong zero-shot accuracy while substantially reducing memory and computational overhead.

### 4.3 Lightweight Fine-tuning Evaluation

We also explored expert fine-tuning to enhance task-specific performance. Although condensation reduces language modeling capacity, fine-tuning can help recover it (Frantar & Alistarh, 2023; Shi et al., 2023; Sanh et al., 2020). Our goal is to show that fine-tuning can restore condensed layers to near-original performance. To this end, we propose a lightweight approach that updates only the condensed layers. For instance, with `CD-MoE-SR`, only 3.8% of parameters require gradients, and SFT on the MMLU training set takes just 2 hours on a single A100 GPU. Following Li et al. (2024a), we fine-tuned using commonsense170k and MMLU training sets.

Table 1 (bottom panel) summarizes the performance gains attributable to lightweight SFT. `CD-MoE-S` improves from an average score of 55.1 to 59.1, while `CD-MoE-SR` jumps from 55.5 to 60.4—nearly matching the original uncondensed performance of 61.6. Notably, ❶ **Commonsense Reasoning:** Task like *BoolQ* shows marked improvements. The `CD-MoE-SR` + SFT lifts performance by up to 7 points in compared to its counterpart before fine-tuning. ❷ **Complex Knowledge Tasks:** On *MMLU*, we observe significant gains (e.g., from 27.1 to 38.5 with `CD-MoE-SR` + SFT), indicating that even after pruning, the model retains foundational knowledge that can be effectively recovered through selective fine-tuning. ❸ **Efficiency:** Despite the notable performance boost, the computational overhead remains low because only a fraction of parameters ($\sim 3\%$) is updated. This design ensures training remains practical for resource-constrained settings, aligning with real-world demands.

For fair comparison, we also apply SFT to Layer Drop, updating only its self-attention sub-layers. While fine-tuning recovers some performance, Layer Drop's complete expert removal fundamentally limits recovery. In contrast, `CD-MoE-SR` retains a condensed expert set, enabling significantly better recovery.

Figure 4 shows how the average accuracy changes after SFT as more layers are condensed. Compared to `CD-MoE-S`, `CD-MoE-SR` delivers consistently higher accuracy with lower memory usage, suggesting that selecting additional routing experts via greedy search preserves more of the model's foundational knowledge. Moreover, `CD-MoE-SR` can restore and even surpass the original model's performance through lightweight SFT, reducing memory usage by up to 20%. These findings indicate that our condensation strategy retains recoverable knowledge despite significant pruning. With targeted lightweight SFT, the condensed experts

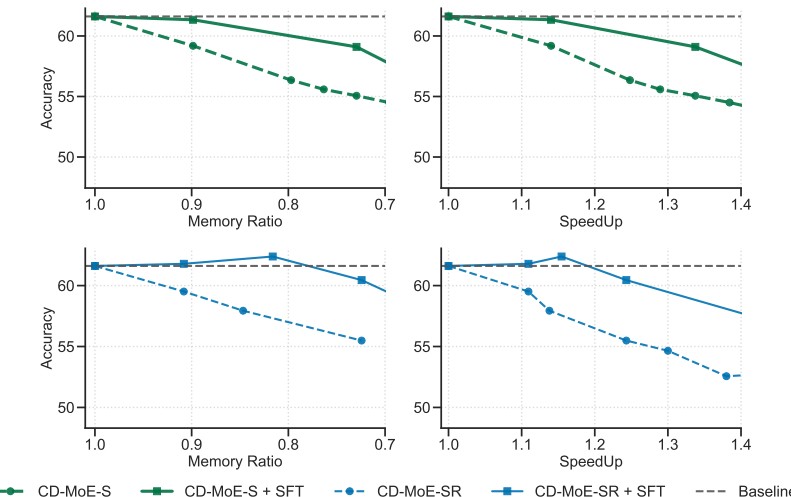

Figure 4: `CD-MoE` with lightweight fine-tuning. Left: SFT results on `CD-MoE-S` with increasing number of condensed layers. Right: SFT results on `CD-MoE-SR` with increasing number of condensed layers. Baseline indicated performance of the dense model.

regain most of their original performance, and in certain tasks, `CD-MoE-SR + SFT` even exceeds the unpruned baseline.

### 4.4 Experiment Analysis

**Expert selection algorithms**. We evaluate several strategies for selecting the most critical experts at each layer. Our primary baselines rely on easily computable statistical properties of the expert parameters:

- *Random*: Randomly select $K$ experts.

- $L_1$: Calculate the sum of the $L_1$ norms of each expert's parameters, sort them, and select the $K$ experts with the lowest $L_1$ norms.

- *PL_Alpha_Hill*: Following heavy-tailed self-regularization (Martin & Mahoney, 2019; 2020; 2021), we measure each layer's spectral density via the Hill estimator (Hill, 1975; Xiao et al., 2023):

$$\texttt{PL\_Alpha\_Hill}_\ell = 1 + k \left/ \sum_{i=1}^{k} \ln\left(\lambda_{n-i+1}^\ell / \lambda_{n-k}^\ell\right), \right. \tag{4}$$

where $\{\lambda_i^\ell\}$ are sorted eigenvalues and $k$ is chosen via the Fix-Finger method Yang et al. (2023). A lower $\texttt{PL\_Alpha\_Hill}_\ell$ indicates a more heavy-tailed layer, so it is less likely to be pruned.

We also experimented with selecting experts having the highest $L_1$-norms, but observed substantially worse performance, potentially because lower norms often indicate better-converged parameters. In Table 5, we compare these methods to our Greedy Search, which iteratively selects experts that minimize the deviation between the post-condensation and pre-condensation outputs of the layer. Notably, none of the statistical baselines (*Random*, $L_1$, or *PL_Alpha_Hill*) outperform Greedy Search. We hypothesize that the latter's layer-level output fidelity objective directly retains the most impactful experts.

We also measure the time cost of our Greedy Search on the *DeepSeekMoE-16B* model and compare it with the exhaustive traversal in (Lu et al., 2024), which enumerates all expert combinations. Each experiment uses 100 C4 calibration samples with maximum sequence length 512 and batch size 256, running on an 80G A100 GPU. As shown in Table 6, the exhaustive approach requires an intractably large number of inferences and is

Table 5: Comparison of expert selection methods. $L_n$ denotes condensing $n$ layers. Each entry is the average accuracy (%) on eight downstream tasks.

| Methods | $L_6$ | $L_9$ | $L_{12}$ | $L_{15}$ |
|---|---|---|---|---|
| Random | 54.6 | 52.7 | 50.1 | 46.8 |
| PL_Alpha_Hill | **56.8** | 52.5 | 50.3 | 47.3 |
| $L_1$ | 56.5 | 53.2 | 50.4 | 47.5 |
| Greedy Search | 56.5 | **55.5** | **52.8** | **49.0** |

Table 6: Time cost comparison for expert and layer selection. '-' indicates that the method does not require the step, '*' denotes estimates.

| Method | Expert Selection | | Layer Selection |
|---|---|---|---|
| | Inferences (Counts) | Time (Hours) | Time (Hours) |
| (Lu et al., 2024) | 75,041,808 | 2500* | - |
| CD-MoE-S | - | - | 0.12 |
| CD-MoE-SR | 369 | 0.13 | 0.15 |

therefore infeasible. By contrast, Greedy Search requires only a few hundred inferences per layer, striking a favorable balance between accuracy preservation and runtime efficiency.

**Layer selection algorithms.** We compare our proposed *Greedy Search (Layer)* method against two baselines that rank layers by measuring the divergence between the layer outputs before and after condensation:

- **Layer Rank:** For each layer $L_i$, we compute the JS divergence $JS(O_{\text{ref}}(L_i), O_{\text{condense}}(L_i))$ between its output before ($O_{\text{ref}}$) and after condensation ($O_{\text{condense}}$). Layers are then sorted by this divergence in ascending order. The layer set $L_{\text{condense}}$ is constructed by selecting the top-$N$ layers with the smallest divergence, i.e., $L_{\text{condense}} = \{L_i \mid \text{Rank}_{JS}(L_i) \leq N\}$.

- **Global Layer Rank:** For each layer $L_i$, we apply condensation and evaluate the JS divergence $JS(O_{\text{ref}}(\text{Model}), O_{\text{condense}}(\text{Model}, L_i))$ between the model's final outputs before condensation ($O_{\text{ref}}$) and after condensing only that layer ($O_{\text{condense}}$). Layers are sorted by this global divergence in ascending order. The layer set $L_{\text{condense}}$ is constructed by selecting the top-$N$ layers with the smallest global divergence, i.e., $L_{\text{condense}} = \{L_i \mid \text{GlobalRank}_{JS}(L_i) \leq N\}$.

Table 7 shows a comparison of these methods in the context of `CD-MoE-SR`. Notably, our Greedy Search consistently surpasses the baselines, underscoring its effectiveness in identifying layers that minimally disrupt the overall model output. We also measure the time needed to condense 15 of the 27 MoE layers using the same setup as in expert selection (see Table 6). This operation requires only 0.15 hours, further validating the computational efficiency of the `CD-MoE` approach.

**Searching metrics.** To quantify how well each condensed model approximates the original model's output, we also compare several metrics during the Greedy Search process. Following Zhang et al. (2024), we adopt *JS* divergence as our primary measure, as it has been shown to be more sensitive to output changes than angular distance or Euclidean distance. We compare *JS* against two alternative metrics:

Table 7: Comparison of layer selection methods. $L_n$ indicates the condensation of $n$ layers. Entries are average accuracies (%) on eight test datasets.

| Methods | $L_6$ | $L_9$ | $L_{12}$ | $L_{15}$ |
|---|---|---|---|---|
| Layer Rank | 45.5 | 44.2 | 42.8 | 42.2 |
| Global Layer Rank | 55.7 | 53.1 | 48.8 | 45.9 |
| Greedy Search | **56.5** | **55.5** | **52.8** | **49.0** |

Table 8: Performance of different metrics for layer selection. $L_n$ indicates the number of condensed layers. The values shown are average accuracies (%) on eight test datasets.

| Metrics | $L_6$ | $L_9$ | $L_{12}$ | $L_{15}$ |
|---|---|---|---|---|
| $PPL$ | **58.7** | 55.4 | 50.9 | 47.2 |
| $KL$ divergence | 57.1 | 54.7 | 52.2 | 48.5 |
| $JS$ divergence | 56.5 | **55.5** | **52.8** | **49.0** |

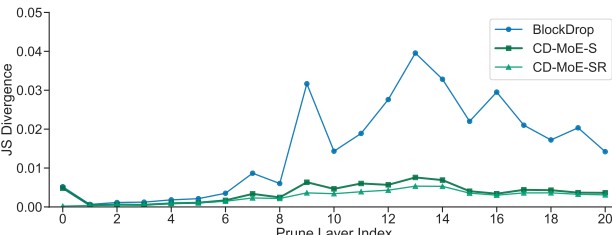

Figure 5: Layer-wise JS curves. The three curves represent the $JS$ divergence between the outputs of a pruned layer and the outputs of the original layer, using different pruning methods.

- *Perplexity (PPL)*: For each calibration sample, we compute the change in PPL and select layers that minimize the average difference in PPL between the original and condensed models.

- *KL Divergence*: We measure the shift between the two output distributions using $KL$ divergence and select layers with minimal $KL$ distance.

Table 8 presents the results. While $PPL$ achieves a marginally better score for $L_6$, $JS$ divergence excels as more layers are condensed ($L_9, L_{12}, L_{15}$). These findings substantiate the choice of $JS$ divergence as a robust metric for maintaining the model's overall quality under heavier compression.

**Explanation of shared expert dominance**. Our analysis of Table 1 reveals a noteworthy finding: `CD-MoE-S`, while retaining only shared experts in condensed layers, achieves comparable zero-shot performance to `CD-MoE-SR`, which preserves both shared and selected routed experts, significantly outperforming *Block Drop*. This suggests that shared experts alone may sufficiently preserve model capability.

To investigate this phenomenon, we quantitatively evaluated layer-wise output deviations between pruned and original models using $JS$ divergence (Figure 5). The results demonstrate that `CD-MoE-S` induces minimal output distortion, closely approximating `CD-MoE-SR`'s behavior. In contrast, *Block Drop*'s aggressive layer removal causes substantially greater output deviation, explaining its inferior performance.

## 5 Conclusion

We have presented `CD-MoE`, a novel framework for compressing large-scale Mixture-of-Experts (MoE) models by selectively condensing them into denser layers. Our approach removes the token routing mechanism and uses a greedy search algorithm to prune the majority of less significant experts, allowing all tokens to activate only a handful of highly impactful experts. Extensive experiments on several fine-grained MoE models demonstrate that `CD-MoE` effectively retains most of the original model's accuracy while achieving significant memory reduction and inference speedup. Moreover, we demonstrate that lightweight expert fine-tuning, focused solely on the condensed layers, can further reclaim the original model's performance within a few hours on a single GPU. These results highlight the practical benefits of `CD-MoE` for resource-constrained scenarios, where memory and computational efficiency are critical. Future directions include exploring the synergy of `CD-MoE` with quantization and knowledge distillation, and extending the method to more diverse MoE structures lacking shared experts.

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

# A   Appendix

## A.1   Implementation Details

We utilize Hugging Face and PyTorch for the implementation of our work [1]. All inference and fine-tuning operations are executed using bf16 (Brain Floating Point 16) precision on NVIDIA A100 GPU equipped with 80GB of memory. During the fine-tuning phase, we employ an initial maximum learning rate of $1 \times 10^{-4}$, implement a warm-up ratio of 10% to gradually ramp up the learning rate, and utilize cosine annealing as the learning rate scheduler to ensure smooth convergence and prevent potential overfitting. Due to the large computational requirements and our limited resources, all experiments were conducted in a single run.

The expert configurations of *DeepSeekMoE-16B*, *Qwen1.5-MoE-A2.7B* and *Qwen2-MoE-57B-A14B* is represented at Table A.1.

Table A.1: Expert configurations in MoE models

| Model | DeepSeekMoE-16B | Qwen1.5-MoE-A2.7B | Qwen2-MoE-57B-A14B |
|---|---|---|---|
| Parameters | 16,375,728,128 | 14,315,784,192 | 57,421,636,608 |
| Activated Parameters | 2,828,650,496 | 2,689,177,536 | 14,249,346,048 |
| Number of Experts | 66 | 64 | 72 |
| Number of Activated Experts | 8 | 8 | 16 |
| Ratio of Activated Experts | 12.1% | 12.5% | 22.2% |
| Number of Shared Experts | 2 | 4 | 8 |
| Ratio of Shared Experts in Activated Experts | 25% | 50% | 50% |

## A.2   Calibration Data Analysis

C4 dataset commonly serves as the calibration benchmark. In our study, we aim to ensure that the performance of the condensed model aligns closely with specific task requirements. To this end, we experiment by utilizing questions extracted from multiple test sets as calibration data, thereby tailoring the language modeling capabilities of the condensed model to the tasks at hand. Specifically, we sampled 20 questions from each of eight distinct test sets, resulting in a total of 160 questions.

Table A.2: Performance comparison across different calibration data across different layer pruning indexes. $L_n$ indicates the condensation of $n$ layers. Entries represent evaluation metrics (%).

| Methods | $L_3$ | $L_6$ | $L_9$ | $L_{12}$ | $L_{15}$ |
|---|---|---|---|---|---|
| DownStream Data | 59.5 | 56.1 | 55.3 | 51.9 | 48.7 |
| Audio Token | 59.0 | **56.7** | 55.0 | 52.0 | **49.1** |
| C4 | **59.5** | 56.5 | **55.5** | **52.8** | 49.0 |

As demonstrated in Fig A.2, it is noteworthy that across varying numbers of condensed layers, the C4 data consistently achieves superior performance compared to the downstream tasks data. This suggests that the diversity and comprehensiveness of the C4 dataset provide a more robust foundation for calibrating the condensed model. Consequently, we adopt C4 as the calibration data in all experiments. This finding is consistent with the work by Bandari et al. (2024), where they perform unstructured pruning of LLM parameters, while we focus on layer condensation.

Additionally, we probed more extreme domain shifts by using small audio-domain samples from the open-source Emilia dataset(He et al., 2024a), processed through wav2vec(Baevski et al., 2020) for calibration.

---

[1]Our repository is built on top of Transformers: https://github.com/huggingface/transformers

The accuracy remained broadly comparable to the C4-based results, as reported for the average accuracy of *DeepSeekMoE-16B* on commonsense reasoning and MMLU. This demonstrates that `CD-MoE` is not highly dependent on a specific calibration dataset.

