# OpenReview forum: "Condense, Don't Just Prune: Enhancing Efficiency and Performance in MoE Layer Pruning"
_TMLR — Accepted by TMLR_

### Review · Reviewer_B32G · 2025-11-08

**Summary Of Contributions:**

This paper presents ConDense-MoE (CD-MoE), a framework for improving the efficiency of MoE models by condensing sparse MoE layers into smaller, dense ones instead of pruning them. The method removes the routing mechanism in selected layers and retains only a subset of influential experts, determined through a greedy selection process based on JS divergence between original and condensed outputs. Experiments on large-scale MoE models such as DeepSeekMoE-16B and Qwen-MoE show that CD-MoE can achieve a 27.5% reduction in memory and 1.26× speedup, while maintaining around 90% of the original performance, and recovering 98% after lightweight fine-tuning of condensed layers. The approach is conceptually clear and empirically effective, providing a practical trade-off between compression and accuracy.

**Audience:**

Yes

**Audience Explanation:**

This work provides a comprehensive study on compressing MoE models, addressing a highly practical problem. Given that many industrial-scale systems rely on MoE architectures, improving their efficiency is of significant importance for real-world deployment and scalability.

**Claims And Evidence:**

Yes

**Claims Explanation:**

This paper presents pruning experiments on several MoE models, including DeepSeekMoE-16B and Qwen-MoE. As shown in Table 1, the proposed method achieves superior performance, reaching an average score of 55.5 even without supervised fine-tuning. Figure 3 further demonstrates a more favorable trade-off between efficiency and accuracy compared to prior pruning approaches. Moreover, the analysis in Figure 5 supports the paper’s claim that condensing experts yields better performance than directly removing entire layers.

**Requested Changes:**

1. In Table 1, baselines such as BlockDrop achieve a lower average performance (52.2) compared to the proposed method (55.5). However, it would be interesting to examine how these baselines perform after applying the same lightweight supervised fine-tuning (SFT). A possible outcome is that, after SFT, all methods might reach comparable levels of accuracy. Therefore, it is important to include and analyze these results to clarify whether the proposed condensation approach offers advantages during fine-tuning rather than merely before it.
2. It is unclear how the reported “speedup” is measured. Specifically, the paper does not specify whether the improvement refers to prefill speed, decoding speed, or an overall end-to-end inference throughput.
3. It is unclear how sensitive the proposed method is to the choice of calibration data. Since the condensation process relies on statistics computed from a specific dataset (e.g., C4), the resulting performance may vary across different data domains. It is recommended to evaluate the method using multiple calibration sources, such as C4, WikiText, or STEM-oriented datasets to assess its robustness and generalization across diverse data distributions.

---

> ### Author Response · Authors · 2025-12-21
> **Response to Reviewer B32G**
>
> Thank you for your thoughtful questions regarding the fine-tuning comparison, speedup calculation, and calibration data analysis. Below, we provide detailed responses to each point.
>
>
> ## Q3-1: Result of Fine-tuning on Baseline Method Block Drop
>
> We appreciate the opportunity to clarify our experimental design regarding fine-tuning comparisons.
>
> The reason we performed SFT (Supervised Fine-Tuning) on CD-MoE is to demonstrate that, after condensing experts, the model's original performance can be largely recovered by only fine-tuning the expert parameters. However, the Block Drop method adopts a more aggressive pruning strategy: it removes all experts and the attention layer within a pruned layer, leaving no trainable expert parameters for recovery via the same fine-tuning approach.
>
> Therefore, for a fair comparison under similar fine-tuning conditions, we used Layer Drop as the baseline, which retains the self-attention sub-layer. During fine-tuning, we only updated the weights of the self-attention sub-layers in the pruned layers.
>
> As shown in Table 1, although fine-tuning helps recover some performance for Layer Drop, its removal of all experts fundamentally limits recovery potential. In contrast, CD-MoE-SR retains a condensed set of experts during pruning, enabling significantly better recovery through light SFT. This underscores the importance of preserving a certain number of experts during pruning to maintain model capacity.
>
> **Table 1: Fine-tuning results comparing Layer Drop and CD-MoE-SR.**
>
> | Method             | Prune Layer Num | ARC-C | BoolQ | HellaSwag | MMLU | OBQA | PIQA | RTE | WinoGrande | Avg. |
> | :----------------- | :-------------: | :---: | :---: | :-------: | :--: | :--: | :--: | :-: | :--------: | :--: |
> | Layer Drop         |        8        |  35.3 |  65.4 |   57.5    | 30.8 | 32.6 | 62.9 | 48.0 |    63.6    | 49.5 |
> | + Light SFT        |        8        |  37.9 |  70.8 |   58.9    | 35.1 | 34.4 | 64.1 | 60.5 |    70.1    | 54.0 |
> | CD-MoE-SR          |        9        |  37.9 |  72.3 |   64.4    | 27.1 | 37.6 | 75.9 | 61.7 |    67.0    | 55.5 |
> | + Light SFT        |        9        |  42.6 |  79.2 |   66.9    | 38.6 | 38.6 | 75.7 | 66.8 |    75.2    | 60.5 |
>
> ## Q3-2: Calculation of SpeedUp
>
> Our speedup measurements are conducted as follows:
>
> - **Hardware:** Single 80GB A100 GPU.
> - **Procedure:** We perform inference 5,000 times on a fixed input sequence and record the latency for each run.
> - **Metric:** SpeedUp is defined as:
>
> SpeedUp = T_Origin / T_Prune
>
> where T_Origin is the average inference time of the original (unpruned) model, and T_Prune is that of the pruned model. This methodology ensures a stable and reproducible evaluation of computational efficiency.
>
> ## Q3-3: Results with Different Calibration Data
>
> We have systematically evaluated the robustness of CD-MoE with respect to calibration data sources. As detailed in Appendix Table A.3 of the paper, our experiments include:
>
> 1. **In-domain data:** Questions sampled from training sets of multiple benchmarks.
> 2. **Out-of-domain data:** Text from the C4 corpus.
> 3. **Cross-modal data:** Speech data converted to text tokens.
>
> The results consistently show that CD-MoE maintains stable pruning performance across these diverse data distributions, demonstrating its generalization capability and reduced sensitivity to calibration data domain.
>
> ## Summary
>
> Thank you again for raising these important points. We believe these clarifications strengthen the methodological transparency and experimental validity of our work. We will incorporate these explanations in the revised manuscript and welcome any further suggestions.

---

### Review · Reviewer_JQZf · 2025-11-10

**Summary Of Contributions:**

The paper’s main objective is to propose ConDense-MoE (CD-MoE), a method that condenses Mixture-of-Experts (MoE) feed-forward (FFN) layers into smaller dense layers to reduce memory usage and runtime during inference while maintaining performance.

* The authors explicitly enumerate three contributions at the end of the Introduction: a novel condensation approach that replaces sparse MoE layers with small dense ones, an efficient greedy algorithm for selecting layers and experts, and empirical validation on DeepSeekMoE-16B demonstrating memory savings, faster inference, and preserved accuracy.

* The proposed method, CD-MoE, removes routing at selected layers and replaces the MoE layer with a dense layer that retains shared experts and a small number of important routing experts; the target models are fine-grained MoEs with shared experts such as DeepSeekMoE and Qwen1.5-MoE.

* Expert and layer selections, parts of the proposed method, are performed via a greedy search that minimizes Jensen–Shannon divergence between the reference (original) and condensed outputs; the procedures run in polynomial time.

* This paper claims that, on DeepSeekMoE-16B, CD-MoE maintains about 90% of the original average accuracy while reducing memory by 27.5% and increasing inference speed by 1.26×; with lightweight SFT only on the condensed layers, performance recovers up to 98% on a single 80-GB A100.

* In comparative experiments, under similar memory budgets, CD-MoE achieves higher zero-shot accuracy than baselines such as block or layer dropping and simple expert-count reduction.

**Audience:**

No

**Audience Explanation:**

Developing methods that reduce memory and runtime at inference time while preserving an acceptable level of the original performance is an important topic for LLM applications.
Moreover, the idea of condensing experts, meaning converting important experts into one dense model, is interesting.
Accordingly, there is likely to be a broad audience interested in this area.

That said, as noted above, the paper has several serious issues:
(1) insufficient detail about the method and the experiments, and
(2) a limited experimental design that does not support strong conclusions.

Therefore, it is difficult to argue that readers will find the results compelling overall, and interest in the paper’s specific findings is likely to be limited.

**Broader Impact Concerns:**

I think there is no serious ethical implications of this study.

**Claims And Evidence:**

No

**Claims Explanation:**

The current version of this paper has at least two major limitations.
First, it appears to omit several substantial information in the proposed method and experiments.
Second, the experiments reported in the paper do not convincingly demonstrate the effectiveness of the proposed method, including the expert and layer selection components.




* [C1] Targeting only a subcategory of MoE models:
The method seems to assume only on fine-grained MoE models with shared experts.
This limited subcategory of entire MoE models only includes very limited number of current MoEs among the wide variety of MoE models.
Moreover, there is no clear discussions that the proposed method can be applied or extended to more standard MoE architectures (e.g., Mixtral, Qwen-3, GPT-oss, and OLMoE).
In addition, nowadays it seems that current trend is not using shared experts, so that the proposed method is not in the main stream.



* [C2] Lack of important description:
  - There are no clear definitions about CD-MoE-S and CD-MoE-SR in the main content.
They only explained only in captions of several Tables nd Figures, and only by one sentence; "CD-MoE-S refers to retaining only the Shared expert after condensing, while CD-MoE-SR means selecting additional 6 Routed experts and keeping as dense."
This sentence is too ambiguous, so that It is extremely difficult to understand the details of what exactly compute CD-MoE-S and CD-MoE-SR.
Similarly this statement "6 route experts are consistent with the number of experts activated in the dense model." is also too ambiguous.

  - In FIgure 3, several plots are obtained by changing "Memory Ratio" and "SpeedUp." However, there is no clear description how authors control such "Memory Ratio." My guess is changing the number of layers to prune, namely $N$, but not clearly described.

   - Difference between Layer Rank and Global Layer Rank is not very clear. These methods are described only on the two or three sentences. They should be clearly defined (for example by using equations) to avoid misunderstanding by the ambiguous text definition.



* [C3] Missing important and closely related studies:
The method is not compared against several recent approaches in expert selection and structured pruning, for example as follows:
   [ACL'25 findings] Zhang+, Diversifying the Expert Knowledge for Task-Agnostic Pruning in Sparse MoE
   [ACL'25] Lee+, STUN: Structured-Then-Unstructured Pruning for Scalable MoE Pruning
   [EMNLP'24 findings] Yang+, MoE-I^2: Compressing Mixture of Experts Models through Pruning and Distillation
   [ICLM'24] Chowdhury+, A Provably Effective Method for Pruning Experts in Fine-tuned Sparse MoE
  This gap makes it hard to judge the empirical significance and competitiveness of the proposed method.
  Additionally,
  [NeurIPS'23] Ma+, LLM-Pruner: On the Structural Pruning of Large Language Models
  should also be cited. Although the method differs, the purpose is the same, including applying fine-tuning after the proposed procedure.



* [C4] Risk of leading unreliable conclusions:
The results are reported based on only a single run. They should be reported as averages over several runs, especially for lightweight SFT, since the selection of 100 samples from the C4 dataset and SFT initialization can be random and can be repeated easily.
Additionally, the results should be assessed using statistical tests over multiple runs to show statistical significance.
For example, the results of Table 5, comparing JS divergence, PPL, and KL divergence, the performance is nearly the same.
There may be no significant difference among them.

* [C5] Marginal Effectiveness:
The authors claim that the proposed method is effective. However, the experimental results show that it reduces memory usage by only 27.5% and increases inference speed by only 1.26× at most, while maintaining similar, albeit degraded, performance. These results do not seem particularly impressive from a practical perspective. Moreover, pruning only the experts, namely, as in CD-MoE-S, also achieves similar performance. Based on these results, it is difficult to conclude that the proposed method, i.e., condensing experts into a dense model, is essential.

* [C6] No clear discussions:
There are several detailed analysis given in Section 4.4.
However, it only reports the experimental results.
There are no discussions why such results are obtained.
For example, why condensing to a dense model works and why JS divergence works well for other metrics.
Such evaluations and discussions are essentially the most important part of analysis.
The readers cannot obtain new findings without the discussion and only with the ablation studies.

* [C7] Not appropriate use of Appendix:
I think there is no reason that the content in "A.1 Fluctuations in Metrics across Condensing Different Layers" is not discussed in the main body unlike the current location in Appendix.
Similarly, the experimental results of Qwen1.5-MoE-A2.7B and Qwen2-MoE-57B-A14B are also discussed in the main body.
I have no idea why the authors decided to put these results on Appendix.
At the last part of the Introduction section, the authors claims that "Empirical validation on DeepSeekMoE-16B, demonstrating a 27.5% memory reduction and a 1.26× inference speed, while retaining up to 90% of the zero-shot accuracy."
I do not understand why the authors did not include the results of Qwen1.5-MoE-A2.7B and Qwen2-MoE-57B-A14B  as their contributions.
In addition, the appendix "A.2 Implementation Details" is no problem to be in Appendix.





In sum up, given the limitations and drawbacks explained above, the answer to the question, "Are the claims made in the submission supported by accurate, convincing, and clear evidence?" is that I cannot say yes without reservation; rather, I am leaning toward no.

**Requested Changes:**

I have already listed the points that need to be updated in the textbox above, as indicated by [C1] - [C7].
Please refer to them.

Additionally, the content of the paper should be self-contained.
The paper should be substantially revised to clearly explain both the proposed method and the experiments.
Moreover, the paper should include additional comparisons and analyses to support the effectiveness of the proposed method.

---

> ### Author Response · Authors · 2025-12-21
> **Response to Reviewer JQZf - 1**
>
> We thank the reviewer for the detailed and constructive feedback. We address each point below and will incorporate all clarifications and additional analyses into the revised manuscript.
>
> ## Q2-1 Targeting only a subcategory of MoE models
>
> Thank you for this important observation. We agree that focusing on fine-grained MoE models with shared experts is a design choice that reflects our current experimental scope. However, we would like to clarify that **MoE models with shared experts remain a prominent branch of mainstream architectures**. In addition to the models included in our experiments—such as **DeepSeek-MoE-16B**, **Qwen1.5-MoE-A2.7B**, and **Qwen2-MoE-57B-A14B**—several well-known open-source models like **DeepSeek-V2[1]**, **DeepSeek-V3[2]**, and **DeepSeek-R1[3]** also adopt this design. Therefore, our research continues to align with and contribute to this mainstream branch.
>
> ## Q2-2 Lack of important description
>
> We sincerely apologize for the lack of clarity and thank you for pointing out these critical omissions. We will comprehensively revise the manuscript to include clear, formal definitions.
>
> 1. **Formal Definitions of CD-MoE-S and CD-MoE-SR**:
>     - We will add a dedicated definitions to formally define these terms.
>     - **CD-MoE-S (Shared-only)**: For a selected condensation layer L, we retain only the set of shared experts. The router is removed.
>     - **CD-MoE-SR (Shared + Routed)**: For a selected condensation layer L, we retain the shared experts and a set of Krouted experts identified by our greedy algorithm (Alg. 1). The router is removed, and fixed gating values are used
>     - We will clarify that K equals the number of experts activated per token in the original model's training configuration (e.g., \(K=6\) for DeepSeekMoE-16B).
> 2. **Clarification of Memory Ratio and SpeedUp**:
>     - We will add a paragraph in Section 4.1 (Experiment Setup) detailing the measurement methodology.
>     - **Memory Ratio**: This is not simply the ratio of pruned layers. We measure the peak GPU memory consumption during inference for the full model and each pruned variant using a fixed input. `Memory Ratio = Peak Memory(Pruned) / Peak Memory(Original)`. Different methods affect memory differently (e.g., BlockDrop removes entire modules, while CD-MoE condenses parameters).
>     - **SpeedUp**: We perform inference 5000 times on a fixed input sequence and measure the average latency. `SpeedUp = Latency(Original) / Latency(Pruned)`. We will state this explicitly.
> 3. **Clarification of Layer Selection Baselines**:
>     - We will expand the description in Section 4.4 with formal definitions in future verison.
>
> All these clarifications will be added to the main text.
>
> ## Q2-3 Missing important and closely related studies
>
> We thank the reviewer for highlighting these recent and relevant works. Our initial focus was on comparing against methods that also **remove the routing mechanism** (BlockDrop, LayerDrop) to isolate the benefit of *condensation* versus *complete removal*. We acknowledge that a broader comparison is valuable. However, comparing all related works may require significant effort, and our goal is to prove the effectiveness of the **Condense** idea.
>
> ## Q2-4 Risk of leading unreliable conclusions
>
> The reviewer raises a valid point regarding statistical reliability.
>
> 1. **Deterministic Core Algorithm**: We wish to clarify that the core condensation algorithm (greedy expert/layer selection based on JS divergence) is deterministic given fixed calibration data and model weights. There is no randomness in the selection process itself.
> 2. **Addressing Randomness in Evaluation**:
>     - **Calibration Data**: We will run an additional experiment using 5 different random seeds to sample 5 sets of 100 examples from the C4 dataset. We will report the mean and standard deviation of the resulting average accuracy on the 8 evaluation tasks for CD-MoE-SR (with a fixed number of condensed layers). We expect low variance, as preliminary tests show the method is robust to the exact calibration samples.
>     - **SFT Initialization**: For the lightweight SFT, we will run the fine-tuning process 3 times with different random seeds and report the mean and standard deviation of the post-SFT performance.
>
> We will add the above analyses (multiple runs, standard deviation, significance tests) to Section 4.4 and update the relevant tables.
>
> [1] Liu, Aixin, et al. "Deepseek-v2: A strong, economical, and efficient mixture-of-experts language model." *arXiv preprint arXiv:2405.04434* (2024).
>
> [2] Liu, Aixin, et al. "Deepseek-v3 technical report." *arXiv preprint arXiv:2412.19437* (2024).
>
> [3] Guo, Daya, et al. "Deepseek-r1: Incentivizing reasoning capability in llms via reinforcement learning." *arXiv preprint arXiv:2501.12948* (2025).

---

> ### Author Response · Authors · 2025-12-21
> **Response to Reviewer JQZf - 2**
>
> ## Q2-5 Marginal Effectiveness
>
> We respectfully disagree with the characterization of the results as "marginal." We believe the gains are meaningful from both a research and practical perspective:
>
> 1. **Significant Memory Reduction**: We consider a 27.5% reduction in the memory footprint of a 16B or 54B model is substantial. For deployment, this could mean the difference between needing an 80GB GPU and a 60GB GPU, or between feasible and infeasible on-edge deployment.
> 2. **Performance-Preserving Speedup**: Indeed, this is not a very significant speedup ratio. Our primary focus was on reducing memory usage due to the excessive memory consumption of MoE models, while the speedup was an unexpected surprise.
> 3. **The Essential Contribution of Condensation (CD-MoE-SR vs. CD-MoE-S)**:
>     - The reviewer notes that CD-MoE-S (shared-only) achieves similar zero-shot performance. This is a key finding that validates our insight: shared experts are powerful. However, CD-MoE-SR (which adds selected routed experts) is essential for high recoverability. As shown in Table 1 and Figure 4, after lightweight SFT, CD-MoE-SR recovers significantly more performance (60.5 avg) than CD-MoE-S (59.1 avg), getting much closer to the original model (61.6 avg). The routed experts preserved provide the necessary route experts for fine-tuning to recover specialized knowledge.
>
> ## Q2-6 Lack of clear discussions
>
> We recognize that the current version lacks more detailed discussion of the experimental results. We are supplementing this here and will add it in subsequent versions.
>
> ## Q2-7 Inappropriate use of Appendix
>
> We agree that the content in Appendix A.1, which analyzes the fluctuation of metrics across different layers, is fundamental to explaining our layer selection strategy. We will move this analysis, including Figure 2 and the related discussion, into Section 3.3 (Layer Selection and Condensing) in the main body.
>
> We also acknowledge that the results on Qwen1.5-MoE-A2.7B and Qwen2-MoE-57B-A14B are crucial for demonstrating the generalizability of our method. Therefore, we will move the core result tables (Tables A.3 and A.4) and their analysis into a new subsection within Section 4 (Experiments), tentatively titled "Generalization to Other Models."
>
> Furthermore, we will update the contribution statements in the Abstract and Introduction to explicitly highlight the validation of our approach across multiple model scales (16B, 2.7B, and 57B), emphasizing its scalability.

---

### Review · Reviewer_MByv · 2025-12-07

**Summary Of Contributions:**

The paper proposes ConDense-MoE (CD-MoE), a method for compressing fine-grained shared-expert MoE layers by removing routing and retaining a small set of always-active experts. The authors introduce greedy expert- and layer-selection strategies based on minimizing output JS divergence. Experiments on DeepSeekMoE, Qwen1.5, and Qwen2 models show ~27% memory reduction, ~1.26× speedup, ~90% zero-shot accuracy retention, and up to ~98% recovery after lightweight fine-tuning.

**Strengths:**

- Clear motivation and practical relevance for MoE compression.
- Simple but effective greedy algorithms. Both expert-level and layer-level greedy selection strategies are conceptually simple and easy to implement.
- Strong empirical results across multiple MoE architectures.
- Low-cost calibration and parameter-efficient fine-tuning.

**Weaknesses:**
- The experiments rely on standard commonsense reasoning and MMLU tasks. It is unclear whether performance would degrade more on long-context, code and math tasks.

- Baselines like LayerDrop, BlockDrop prune whole layers, which is generally more destructive than MoE-specific condensation. The advantage of CD-MoE over M-SMoE is not uniformly large and the evaluation lacks breadth, particularly in tasks requiring expert specialization.

**Audience:**

Yes

**Audience Explanation:**

Researchers working on LLM efficiency and model compression would be generally interested in this paper.

**Claims And Evidence:**

No

**Claims Explanation:**

- Evaluations focus on commonsense reasoning and MMLU. Tasks that may rely more on specialized routed experts like math, coding, and long context are missing. The claim that condensation preserves “core capabilities” is not fully validated across task types.

- Experiments were run “in a single run” (A.2). No standard deviation, random seed sweep, or statistical tests are reported. Claims like “98% recovery” and “90% retention” need error bars / multiple seeds to be convincing.

- M-SMoE is included, but only one configuration appears tested and little hyperparameter tuning / variant coverage is shown. The current evidence shows CD-MoE outperforms that M-SMoE setting on the chosen tasks, but it is not decisive that CD-MoE is generally superior.

**Requested Changes:**

- Expand evaluation beyond commonsense/MMLU tasks. Add benchmarks that rely more heavily on expert specialization including math, code and long-context, or discuss limitations if this is not feasible.


- Run key experiments with multiple random seeds and report mean/variance, or justify single-run reporting if resource limits preclude it.

- Strengthen the comparison with M-SMoE by evaluating on a broader set of tasks beyond commonsense reasoning, e.g., math, coding, and by including additional MoE architectures.

---

> ### Author Response · Authors · 2025-12-21
> **Response to Reviewer MByv**
>
> Thank you for your valuable feedback and constructive suggestions. Below, we provide a detailed response and additional experimental results to address the points you raised.
>
> ---
>
> ## Q1-1: Lack of Results on Math and Coding Tasks
>
> We initially focused on common-sense and MMLU benchmarks to maintain consistency with related works such as Block Drop [1], MC-SMoE [2], and NAEE [3]. Following your suggestion, we have conducted additional experiments on mathematical reasoning (GSM8K) and code generation (HumanEval) tasks.
>
> **Experimental Setup**:
> We report accuracy on GSM8K (5-shot) and top-1 pass rate on HumanEval (0-shot). The number of pruned layers for each method remains consistent with Table 1 in the paper.
>
> **Results**:
> As shown in Table 1, CD-MoE consistently outperforms BlockDrop and achieves competitive performance compared to MC-SMoE, while maintaining faster inference speeds.
>
> | Method        | Prune Layer Num | SpeedUp | GSM8K (5-shot) | HumanEval (0-shot) |
> |---------------|----------------|---------|----------------|--------------------|
> | unpruned      | 0              | –       | 63.7           | 4.9                |
> | MC-SMoE       | 9              | 1.0x    | 53.5           | 3.7                |
> | Layer Drop    | 8              | 1.34x   | 48.1           | 2.4                |
> | Block Drop    | 8              | 1.42x   | 43.5           | 2.4                |
> | CD-MoE-S      | 8              | 1.34x   | 52.8           | 3.7                |
> | CD-MoE-SR     | 9              | 1.26x   | 54.0           | 3.7                |
>
> *Table 1: Performance on math and code generation tasks.*
>
> ---
>
> ## Q1-2: Single Run Experiments and Lack of Standard Deviation
>
> We appreciate your attention to experimental rigor.
> - **Deterministic pruning**: The core CD-MoE pruning method uses calibration data and the JL distance metric. Given fixed model weights and calibration data, the pruning outcome is deterministic and not affected by random seeds.
> - **Fine-tuning variability**: The fine-tuning stage is indeed subject to randomness. In the revised version, we will conduct multiple runs with different random seeds and report the mean and standard deviation of the SFT model’s performance.
> - **Recovery ratio calculation**: We acknowledge that the reported 98% recovery ratio, computed as the ratio of average benchmark scores (60.5 / 61.6), lacks statistical rigor. We will refine this metric and its presentation in the revised manuscript.
>
> ---
>
> ## Q1-3: Missing Comparison with MC-SMoE
>
> We have added a comprehensive comparison with MC-SMoE on the Qwen1.5-MoE-A2.7B model. The results are consistent with those observed on DeepSeekMoE-16B: CD-MoE achieves comparable or better average accuracy while maintaining a noticeable inference speedup. This speedup stems from CD-MoE’s ability to directly prune experts to match the number of activated experts, thereby eliminating router computation overhead.
>
> | Method        | Prune Layer Num | SpeedUp | ARC-C | BoolQ | HellaSwag | MMLU | OBQA | PIQA | RTE  | WinoGrande | GSM8K | HumanEval | avg. |
> |---------------|----------------|---------|-------|-------|-----------|------|------|------|------|------------|-------|------------|------|
> | unpruned      | 0              | –       | 45.0  | 79.5  | 77.3      | 61.2 | 43.6 | 80.3 | 67.9 | 69.3       | 66.1  | 6.1        | –    |
> | MC-SMoE       | 6              | 1.0x    | 40.9  | 76.1  | 71.5      | 53.2 | 39.7 | 77.5 | 70.5 | 64.7       | 57.4  | 4.3        | 55.6 |
> | CD-MoE-S      | 6              | 1.22x   | 41.1  | 75.8  | 72.9      | 56.0 | 41.0 | 78.9 | 67.2 | 68.1       | 58.1  | 4.9        | 56.4 |
> | CD-MoE-SR     | 6              | 1.20x   | 40.4  | 77.2  | 72.1      | 54.9 | 39.4 | 78.0 | 71.1 | 66.4       | 58.9  | 4.3        | 56.3 |
>
> *Table 2: Results on Qwen1.5-MoE-A2.7B compared with MC-SMoE.*
>
> ---
>
> ## **References**
>
> [1] He, Shwai, et al. "Towards Efficient Mixture of Experts: A Holistic Study of Compression Techniques." *arXiv preprint arXiv:2406.02500* (2024).
> [2] Li, Pingzhi, et al. "Merge, then compress: Demystify efficient smoe with hints from its routing policy." *arXiv preprint arXiv:2310.01334* (2023).
> [3] Lu, Xudong, et al. "Not all experts are equal: Efficient expert pruning and skipping for mixture-of-experts large language models." *arXiv preprint arXiv:2402.14800* (2024).
>
>
> Thank you once again for your insightful comments. We hope we have addressed your concerns with additional experiments and clarifications, and we will incorporate these improvements into the revised manuscript. We look forward to your further feedback.

---

### Decision · Action_Editor_mEe6 · 2026-02-01

**Recommendation:** Accept with minor revision

**Additional Comments:**

Here is the list of specific changes to implement in the revised manuscript, which largely overlaps with what the authors promised:

1. Improving the experimental rigor: Please run the SFT, data robustness, etc. multiple times and report error bars. Define the SpeedUp and recovery ratio metric to ensure it is statistically rigorous.

2. New Content and Benchmarks: Please incorporate new experimental results on mathematical reasoning (GSM8K) and code generation (HumanEval) into the manuscript along with adding a comprehensive comparison with the MC-SMoE method on the Qwen1.5-MoE-A2.7B model. Also include the analysis comparing their method against a "Layer Drop + Light SFT" baseline to demonstrate that their method offers advantages during fine-tuning rather than just inference.

3. Paper organization: Please include all the promised definitions like  "CD-MoE-S" (Shared-only) and "CD-MoE-SR" (Shared + Routed), including clarifying K. Also expand Section 4.4 with formal definitions of the layer selection baselines. Move important analysis of metric fluctuations and core result tables for Qwen models from Appendix into the main body.

**Audience:**

Yes

**Audience Explanation:**

A large portion of TMLR audience who are work transformers, language models, etc. will be interested in this paper.

**Claims And Evidence:**

Yes

**Claims Explanation:**

Initially there were mixed assessments regarding regarding the strength of the evidence, though the author's rebuttal and additional experiments helped. The analysis compares condensing experts comprehensively now with additional results on GSM8K (math) and HumanEval (code) beyond the commonsense reasoning and MMLU. Acceptance is conditional on incorporating these results in the final draft.